

# Trophic niche differentiation and utilisation of food resources in Collembola is altered by rainforest conversion to plantation systems

Winda Ika Susanti[1,2], Rahayu Widyastuti[2], Stefan Scheu[1,3] and Anton Potapov[1,4]

[1] J.F. Blumenbach Institute of Zoology and Anthropology, Georg-August Universität Göttingen, Göttingen, Germany
[2] Department of Soil Science and Land Resources, Bogor Institute of Agriculture, Bogor, Indonesia
[3] Center of Biodiversity and Sustainable Land Use, Göttingen, Germany
[4] A.N. Severtsov Institute of Ecology and Evolution, Russian Academy of Science, Moscow, Rusia

Corresponding author
Winda Ika Susanti,
winda.ika-susanti@biologie.uni-goettingen.de

## ABSTRACT

Intensively managed monoculture plantations are increasingly replacing natural forests across the tropics resulting in changes in ecological niches of species and communities, and in ecosystem functioning. Collembola are among the most abundant arthropods inhabiting the belowground system sensitively responding to changes in vegetation and soil conditions. However, most studies on the response of Collembola to land-use change were conducted in temperate ecosystems and focused on shifts in community composition or morphological traits, while parameters more closely linked to ecosystem functioning, such as trophic niches, received little attention. Here, we used stable isotope analysis ($^{13}C$ and $^{15}N$) to investigate changes in the trophic structure and use of food resources by Collembola in Jambi province (Sumatra, Indonesia), a region that experienced strong deforestation in the last decades. Isotopic values of Collembola from 32 sites representing four land-use systems were analyzed (rainforest, rubber agroforest, rubber (*Hevea brasiliansis*) and oil palm (*Elaeis guineensis*) monoculture plantations). Across Collembola species $\Delta^{13}C$ values were highest in rainforest suggesting more pronounced processing of litter resources by microorganisms and consumption of these microorganisms by Collembola in this system. Lower $\Delta^{13}C$ values, but high $\Delta^{13}C$ variation in Collembola in oil palm plantations indicated that Collembola shifted towards herbivory and used more variable resources in this system. Small range in $\Delta^{15}N$ values in Collembola species in monoculture plantations in comparison to rainforest indicated that conversion of rainforest into plantations is associated with simplification in the trophic structure of Collembola communities. This was further confirmed by generally lower isotopic niche differentiation among species in plantations. Across the studied ecosystems, atmobiotic species (Symphypleona and Paronellidae) occupied the lowest, whereas euedaphic Collembola species occupied the highest trophic position, resembling patterns in temperate forests. Some species of Paronellidae in rainforest and jungle rubber had $\Delta^{15}N$ values below those of leaf litter suggesting algivory (*Salina* sp.1, *Callyntrura* sp.1 and *Lepidonella* sp.1), while a dominant species, *Pseudosinella* sp.1, had the highest $\Delta^{15}N$ values in most of the land-use systems suggesting that this species at least in part lives as predator or scavenger. Overall, the results suggest that rainforest conversion into plantation systems is associated with marked shifts in the structure of

trophic niches in soil and litter Collembola with potential consequences for ecosystem functioning and food-web stability.

## INTRODUCTION

Agricultural intensification in Indonesia is associated with deforestation which increased strongly in the last 30 years (*Koh & Ghazoul, 2010*; *Gatto, Wollni & Qaim, 2015*). Large parts of rainforest in lowland Sumatra (Indonesia) have been converted into oil palm (*Elaeis guineensis*) (16% of total area) and rubber plantations (*Hevea brasiliansis*) (12%) (*Gatto, Wollni & Qaim, 2015*). These processes are driven by the high global demand for agricultural products, and positively affect income and employment of local smallholders (*Grass, Kubitza & Krishna, 2020*; *Qaim et al., 2020*). At the same time, conversion of tropical rainforest into plantation systems is associated with major changes in ecological niches of animal species, loss of biodiversity, and thereby with changes in ecosystem functioning (*Barnes et al., 2014*; *Clough et al., 2016*; *Fitzherbert et al., 2008*; *Gilbert, 2012*). These changes affect both the above- and belowground system. Complex and diverse microbial and animal communities in soil regulate important ecosystem functions and support aboveground life (*Bardgett & Van Der Putten, 2014*), but knowledge on effects of land-use change on soil life in the tropics is very limited.

Studies from Sumatra showed that conversion of rainforest into oil palm and rubber plantations is associated with a decline in species diversity, population density, biomass and energy flux in litter macroinvertebrate communities by approximately 45% (*Drescher et al., 2016*; *Grass, Kubitza & Krishna, 2020*). Uneven decline in energy flux across size classes and trophic levels was documented for meso- and macrofauna soil communities leading to strong alterations in soil food-web structure (*Potapov, Tiunov & Scheu, 2019*). Different basal resources available in different land-use systems result in changes in trophic niches of decomposer and predatory soil invertebrates, and results in reduced abundance of primary decomposers and in soil animals shifting their feeding habits towards herbivory (*Klarner et al., 2017*; *Krause et al., 2019*; *Susanti et al., 2019*). Land-use change may also result in reduced trophic niche differentiation among species in belowground communities (*Korotkevich et al., 2018*), but this has not been investigated in tropical ecosystems.

Collembola are among the most abundant soil decomposer invertebrates, inhabiting various organic substrates and using a wide range of food resources (*Rusek, 1998*). Early studies on food resources of Collembola concluded that the majority of euedaphic and hemiedaphic species feed unselectively on a wide variety of food materials (*Hopkin, 1997*). However, stable isotope analysis showed pronounced trophic niche differentiation among Collembola species in temperate forests (*Chahartaghi et al., 2005*). This differentiation in large has been explained by the taxonomic identity and life forms of Collembola (*Potapov et al., 2016*). Species, living aboveground and on the litter surface (atmobiotic

and epedaphic life forms) are mainly phycophages, feeding on lichens, algae and pollen. Species, living in the litter (hemiedaphic life form) are detritivores feeding on saprotrophic microorganisms and litter. Species, living in soil (euedaphic life form) feed on soil organic matter, roots and fungi (*Ponge, 2000*; *Potapov et al., 2016*). Further, food resources and trophic levels also vary among high-rank taxa with e.g., Poduromorpha occupying higher trophic positions than Entomobryomorpha and Symphypleona, suggesting evolutionary selection for microbivory in the former (*Potapov et al., 2016*). Since lowland tropical forest ecosystems often have a less pronounced organic layer and different plant and animal community composition than temperate forest ecosystems (*Petersen & Luxton, 1982*), they provide different ecological niches for Collembola, potentially resulting in a different trophic structure of communities. To date, information on food resources and trophic niches of Collembola is based on studies from temperate ecosystems, whereas information from tropical ecosystems is virtually lacking entirely.

Over the last two decades, stable isotope analysis has become the most commonly used tool to assess trophic niches of soil animals (*Potapov, Tiunov & Scheu, 2019*). Two isotopic ratios, $^{13}C/^{12}C$ (i.e., $\delta^{13}C$ values) and $^{15}N/^{14}N$ (i.e., $\delta^{15}N$ values), typically are used in food-web studies. Trophic positions and length of trophic chains can be assessed using $\delta^{15}N$ values, whereas the range of $\delta^{13}C$ values reflects variability in the use of basal resources (*Potapov, Tiunov & Scheu, 2019*). Stable isotope composition of consumers follows that in food resources, thus allowing to reveal potential diet switching with land-use change (*Klarner et al., 2017*; *Krause et al., 2019*; *Susanti et al., 2019*).

Here, we use stable isotope analysis to investigate trophic positions and food resources of soil and litter Collembola in four different land-use systems in Sumatra, Indonesia: rainforest, rubber agroforest ('jungle rubber'), and monoculture rubber and oil palm plantations. The study aimed at investigating how trophic positions and food resources of Collembola change after rainforest conversion into agricultural plantations, such as rubber and oil palm, and for the first time, at exploring patterns in trophic niche differentiation among tropical Collembola species. In more detail we investigated the following hypotheses:

1. Analogous to other soil invertebrates, Collembola shift their trophic niches towards herbivory in plantation systems in comparison to rainforest.
2. Due to reduced food resources (poor litter layer), the trophic niche width of Collembola is narrower in plantations in comparison to rainforest.
3. Trophic niche differentiation among Collembola species is more pronounced in rainforest than in plantation systems. Trophic niche differentiation among families and life forms of Collembola in tropical ecosystems follows similar patterns as in ecosystems of the temperate zone.

## MATERIAL AND METHODS

### Site description

Four land-use systems were investigated: lowland rainforest, jungle rubber, rubber and oil palm plantations, located in Jambi province, southwest Sumatra, Indonesia. The study sites were located at a similar altitude varying between 50 and 100 m a.s.l. in two landscapes,

the Harapan and Bukit Duabelas landscape; each land-use system was replicated four times per landscape, resulting in a total of 32 sites (for more details see (*Drescher et al., 2016*)). Lowland rainforest was used as reference, but represents secondary rainforest, which has been logged once by taking out large trees some 30 years ago. Jungle rubber represents a rubber agroforest system originating from rainforest enriched with rubber trees; the age of rubber trees varied between 15–40 years (*Kotowska et al., 2015*). Rubber and oil palm plantations were intensively managed monocultures of an average age of 7 to 16 and 8 to 15 years, respectively (*Drescher et al., 2016*), and were established after logging, clearing, and burning of either rainforest or jungle rubber. Soils at the Harapan landscape are loam Acrisols of low fertility, whereas in Bukit Duabelas the major soil type is clay Acrisol (*Allen et al., 2015*; *Kotowska et al., 2015*). Management practices in these smallholder monoculture plantations are described in more detail in *Allen et al. (2015)*. Oil palm plantations typically were fertilized once in the rainy season and once in the dry season. Typically, 300–500 kg NPK complete fertilizer, 300 kg KCl and 138 kg urea ($CO(NH_2)_2$) were added per hectare and year. Rubber and oil palm plantations were weeded manually or chemically throughout the year. The most commonly used herbicides were paraquat and glyphosate; these were applied at an average rate of 2 to 5 L ha$^{-1}$ y$^{-1}$ (*Allen et al., 2015*; *Clough et al., 2016*; *Kotowska et al., 2015*).

## Sampling procedure

Samples were taken in October 2013 in three 5 × 5 m subplots within 50 × 50 m plots established at each study site (*Drescher et al., 2016*). In each subplot soil samples of 16 × 16 cm were taken including the litter layer and the underlying top soil to a depth of five cm. Animals from both layers were pooled for stable isotope analysis to obtain sufficient amount of animal tissue for the analyses. Animals were extracted by heat (*Kempson, Lloyd & Gheraldi, 1963*) until the substrate was completely dry (6–8 days) using glycerol: water mixture at a ratio of 1: 1 as collection solution. Field collection was conducted under the research permit No. 389/SIP/FRP/SM/X/2013 issued by the State Ministry of Research and Technology of Indonesia (RISTEK) with collection permit No. S.07/KKH-2/2013 issued by the Ministry of Forestry (PHKA) and support from the following persons and organizations who granting us to access and use their properties: village leaders, local plot owners, PT Humusindo, PT Perkebunan Nusantara VI, Harapan Rainforest, and Bukit Duabelas National Park.

## Species identification

Collembola were sorted in Petri dishes using a dissecting microscope. For species-level identification, selected individuals were subsequently cleared in Nesbitt solution and mounted on slides with Hoyer solution. Collembola were identified under a compound light microscope at 400× magnification. The checklist and keys for Indonesian Collembola by *Suhardjono, Deharveng & Bedos (2012)* were used along with publications on Southeast Asian Collembola. Due to a relatively poorly described fauna, in many cases we had to assign individuals to morphospecies without Linnaean names (in total 72% of all identified species); for simplicity, we refer to both as 'species'. When possible, juvenile specimens were

ascribed to species of adults or subadults present in the same sample or in samples from the same plot. After identification, all data on Collembola species and their identification characters were uploaded to Ecotaxonomy database (http://ecotaxonomy.org). In total 56 species from 13 families and 27 genera were found.

## Bulk stable isotope analyses

Stable isotope ratios were- measured from dominant species representing at least 70% of the individuals on each plot (Table S1). A number of rare species were observed only on few sites and such data would be not suitable for a proper analysis of the species and land-use effects and also for analytical facilities in the laboratory. Dominant species were chosen for each plot separately to represent the local 'functional community'. This selection procedure resulted in a total of 30 out of 56 species being included in the analysis across all land-use systems. For stable isotope measurements appropriate amounts of animal tissue (ranging from 0.003 to 1.268 mg) were transferred into tin capsules and dried at 60 °C for 24 h, weighed and stored in a desiccator until analysis. Stable isotope ratios, and total C and N concentration were determined using a coupled system consisting of an elemental analyzer (Eurovector, Milano, Italy) equipped with a Blisotec autosampler (Blisotec, Jülich, Germany) and a Thermo Delta Vplus isotope ratio mass spectrometer connected via a Conflo IV interface (both from Thermo Fisher Scientific, Bremen, Germany) located at the Centre for Stable Isotope Research and Analysis, Göttingen, Germany (Langel & Dyckmans, 2014). Isotope signatures were expressed using the $\delta$ notation as $\delta X(‰) = (R_{sample} - R_{standard})/R_{standard}$, with X representing the target isotope and R the ratio of heavy to light isotope ($^{13}C/^{12}C$ or $^{15}N/^{14}N$). For $\delta^{15}N$ and $\delta^{13}C$ analyses, N in atmospheric air and Vienna Pee Dee Belemnite served as standards, respectively. We use IAEA CH6 ($-10.43‰$, Sucrose) and IAEA 600 ($-27.7‰$, Caffein) as C standards, and IAEA N1 ($0.4‰$) and IAEA N2 ($20.3‰$) for N (both are Ammonium sulfates) for internal calibration.

## Statistical analyses

To compensate for inter-site variation in the isotopic baseline, prior to the analysis all data were normalized to the local leaf litter using the following equations (*Potapov, Tiunov & Scheu, 2019*): $\Delta^{13}C = \delta^{13}C_{Collembola} - \delta^{13}C_{litter}$

$\Delta^{15}N = \delta^{15}N_{Collembola} - \delta^{15}N_{litter}$

Stable isotope values of litter were taken from *Klarner et al. (2017)* who investigated the same sampling sites.

Statistical analyses were performed using R v 3.5.2 (*R Core Team, 2018*) with R studio interface (R Studio, Inc.). First, we analyzed the effect of land-use system on the isotopic composition of Collembola at the community level. Effects of land-use system on the $\Delta^{13}C$ and $\Delta^{15}N$ values of all measured Collembola individuals were tested using Linear Mixed Effect Models (LMM) with species identity as random effect. Species identity was coded as random effect for the following reasons: (1) In this first analysis we were not interested in variations in $\Delta^{13}C$ and $\Delta^{15}N$ values among species but still wanted to account for it, (2) species presence was uneven across plots and land-use systems preventing the analysis of land-use effects, and (3) including species as fixed factor would have compromised the

analysis of land-use effects on $\Delta^{13}$C and $\Delta^{15}$N values of Collembola by reducing error degrees of freedom. The analysis was conducted using the *lmer* function in the *lme4* package (*Bates et al., 2015*). Significance of fixed effects (factors) was tested using the *Anova* function in the *car* package. Significant differences in stable isotope values between land-use systems were tested using the *tukey te* st in the *emmeans* and *multicomp* packages. We also analyzed the effect of land use only for one dominant species, present in sufficient replicates in each land-use system (*Pseudosinella* sp.1) using analysis of variance implemented in the *aov* function. Additionally, the ranges in $\Delta^{13}$C and $\Delta^{15}$N values in each land-use system were calculated as difference between minimum and maximum values and visualized using the Kernel density estimation in the *ggplot 2* package using the *geom_violin* function.

Second, we assessed trophic niche differentiation among species by assessing the effect of species identity on stable isotope composition of Collembola with LMM. We used species identity as factor and either $\Delta^{13}$C or $\Delta^{15}$N values as response variables; sampling plot was included as random effect to account for site-specific differences in trophic niches. The analysis was done separately for each land-use system (eight analyses in total). Only species allowing more than three measurements per land-use system were included.

Third, we analyzed the effects of family identity and life form on stable isotope composition of Collembola using LMM. Here, we used both plot and land-use system as random effects. Significant differences between families and life forms were tested using Tukey contrasts as implemented in the *glth* function in the *multicomp* package. To display the isotopic niche space of Collembola species, family, and life form, ellipses denoting 60% intervals were plotted using the *standard.ellipse* function in the *siar* package, and visualized using the *ggplot* and *ggrepel* packages. We also analyzed interactions between land-use system and life form and between land-use system and family with plot as random effect. In the analyses, we excluded Neanuridae and Onychiuridae as they were only represented by a single species not present in each of the land-use systems. In addition, we also analyzed R square values between species, family, and life form to identify the most important factor. Here, we used plot as random effect. The analysis was done using the *r.squaredGLMM* function in the *lme4* and *MuMin* packages.

## RESULTS

### Community-level changes with land use

In rainforest, $\Delta^{13}$C values of Collembola ranged between 1.5 and 6.0‰ (total range 4.5‰), and in oil palm plantations between −1.5 and 8.5‰ (10.0‰), indicating a wider range of $\Delta^{13}$C values in food resources in the latter. In jungle rubber and rubber plantations, the range of $\Delta^{13}$C values was similar to rainforest (between 1.0 and 6.0 ‰). Mean $\Delta^{13}$C values in rainforest were significantly higher than in plantation systems ($F_{3,146} = 9.90$, $p = 0.001$; Fig. 1A, Tables S2, S3). All $\delta^{13}$C values of Collembola species exceeded those of leaf litter except for one individual of *Isotomiella* cf. *minor* in oil palm plantations (Fig. 2; for details see Tables S6–S9).

In contrast to $\Delta^{13}$C, the range of $\Delta^{15}$N values was largest in rainforest (−5.0 to 19.0‰), lowest in oil palm plantations (−1.0 to 8.0‰), and intermediate in rubber plantations
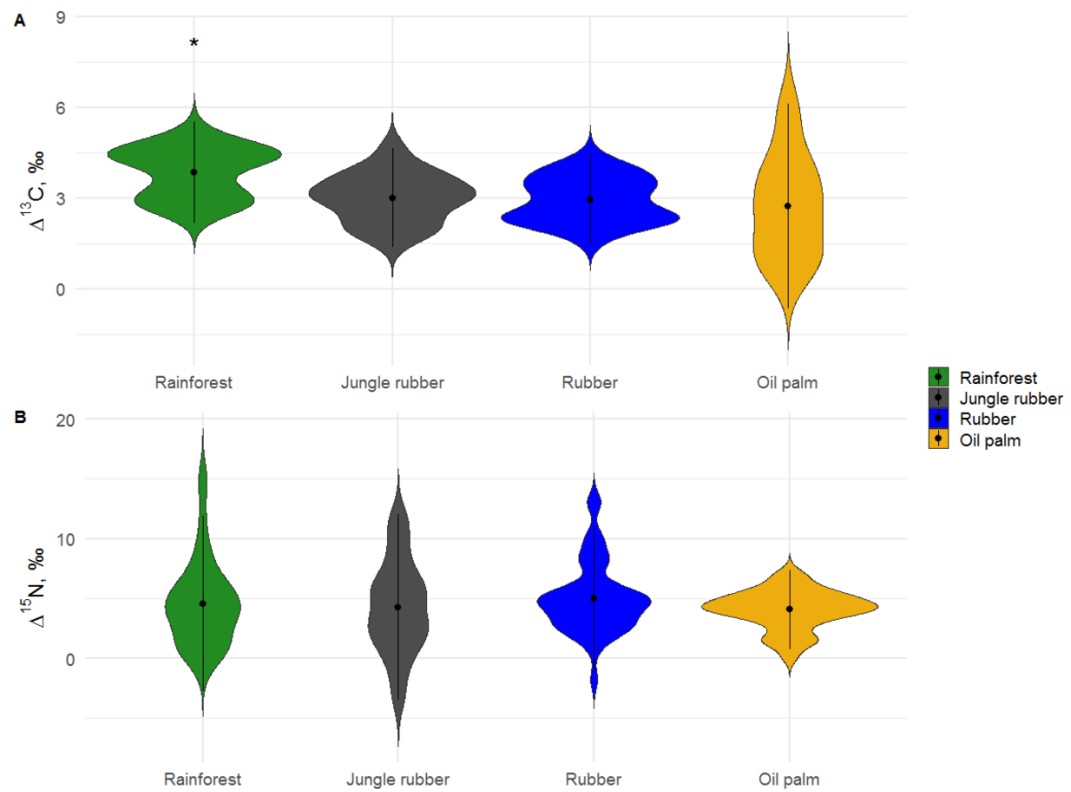

**Figure 1** **Variations in $\Delta^{13}$C and $\Delta^{15}$N values of Collembola among the studied land-use systems (rainforest, jungle rubber, rubber and oil palm plantations).** Variations in $\Delta^{13}$C and $\Delta^{15}$N values of Collembola among the studied land-use systems: (A) rainforest, (B) jungle rubber, (C) rubber and (D) oil palm plantations. Violin plots show frequency distribution of values (mirrored Kernel density estimation), all individual measurements are displayed together independent of taxonomic identity. *Average $\Delta^{13}$C values in rainforest were significantly higher than in the other three land-use systems ($P < 0.05$).

($-5.0$ to $15.0$‰) and in jungle rubber ($-7.5$ to $16.0$‰). Mean $\Delta^{15}$N values of Collembola did not vary significantly among land-use systems ($F_{3,153} = 1.46$, $p = 0.228$; Fig. 1B, Tables S4, S5). Stable isotope values of the most abundant Collembola species present in all land-use systems, *Pseudosinella* sp.1, did not vary significantly among land-use systems both in $\Delta^{13}$C ($F_{3,21} = 1.44$, $p = 0.260$) and $\Delta^{15}$N ($F_{3,21} = 0.88$, $p = 0.467$).

### Niche differentiation among species

$\Delta^{13}$C values across abundant species (represented by at least three measurements, see Methods) varied significantly in rainforest ($F_{4,12} = 6.34$, $p = 0.005$) and jungle rubber ($F_{5,17} = 3.67$, $p = 0.020$), but not in rubber ($F_{5,19} = 1.39$, $p = 0.270$) and oil palm plantations ($F_{2,9} = 1.10$, $p = 0.370$). In rainforest, $\Delta^{13}$C values of *Pseudosinella* sp.1 were highest and differed significantly from those of *Lepidocyrtus* sp.1 and *Pararrhopalites* sp.1, whereas in jungle rubber they were highest in *Homidia cingula* and differed significantly from *Callyntrura* sp.1.

$\Delta^{15}$N values across abundant species varied significantly in rainforest ($F_{4,14} = 5.00$, $p = 0.010$), jungle rubber ($F_{5,18} = 4.70$, $p = 0.006$) and oil palm plantations ($F_{2,9} = 23.59$,

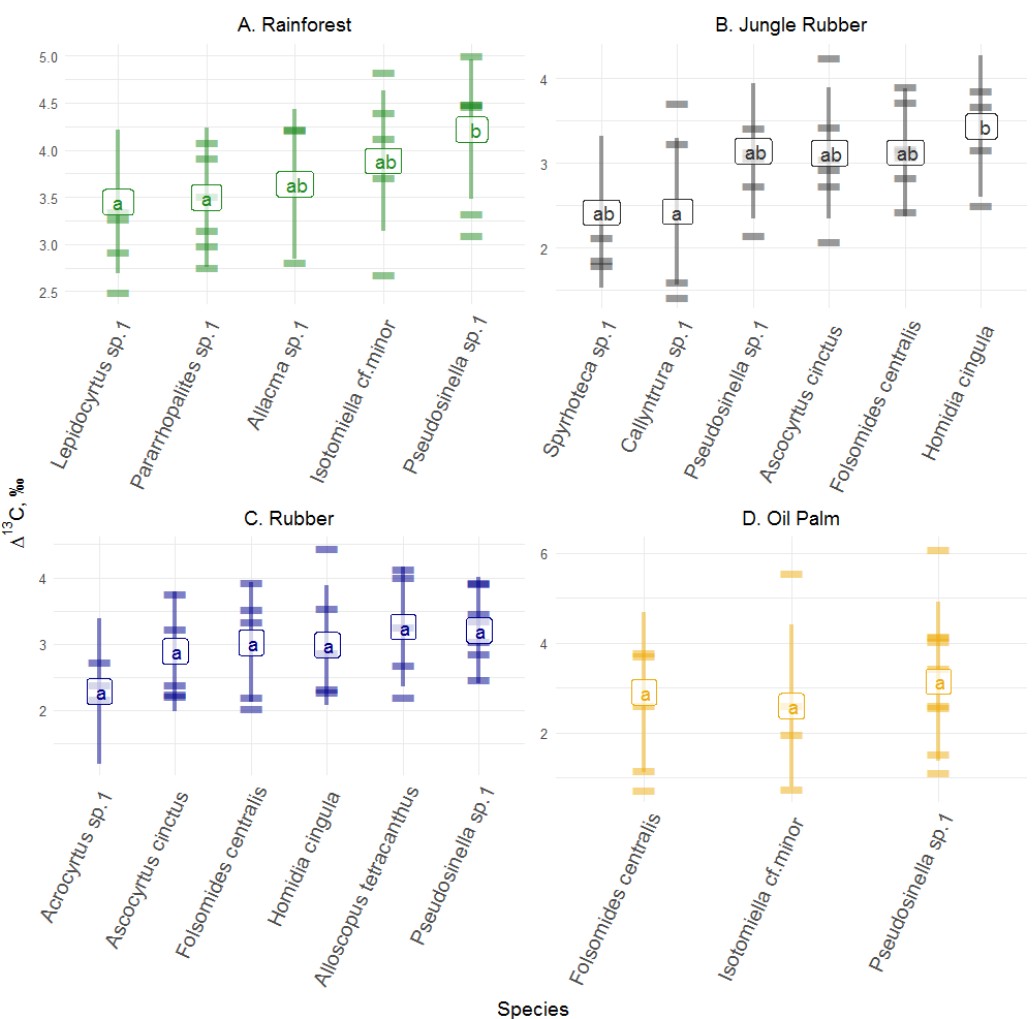

**Figure 2 Differences of $\Delta^{13}$C values of Collembola species in different land-use systems (rainforest, jungle rubber, rubber and oil palm plantations).** Differences of $\Delta^{13}$C values of Collembola species in different land-use systems: (A) rainforest, (B) jungle rubber, (C) rubber and (D) oil palm plantations; model-estimated means (lsmeans) with standard deviation. Horizontal stripes represent individual measurements. Only species with more than three replicates per land-use system were analyzed (see Methods). Isotope values of each species in each land-use system with the same letter are not significantly different according to Tukey's honestly significant difference test ($p > 0.05$).

$p = 0.001$), but not in rubber plantations ($F_{5,19} = 2.43$, $p = 0.072$) (Fig. 3; for details see Tables S10–S13). In rainforest, *Pseudosinella* sp.1 occupied the highest trophic position, followed by *Isotomiella* cf. *minor*, whereas *Allacma* sp.1 occupied the lowest trophic position. Similar to rainforest, in jungle rubber, *Pseudosinella* sp.1 also occupied the highest trophic position, but $\Delta^{15}$N values were lowest in *Callyntrura* sp.1. Overall, *Pseudosinella* sp.1, the most dominant species, occupied the highest trophic position across all species. Across all samples analyzed (20 species in rainforest, 17 species jungle rubber, 15 species in rubber plantations, and 13 species in oil palm plantations) 3% of the measurements had $\Delta^{15}$N values below that of litter (see Fig. S1).

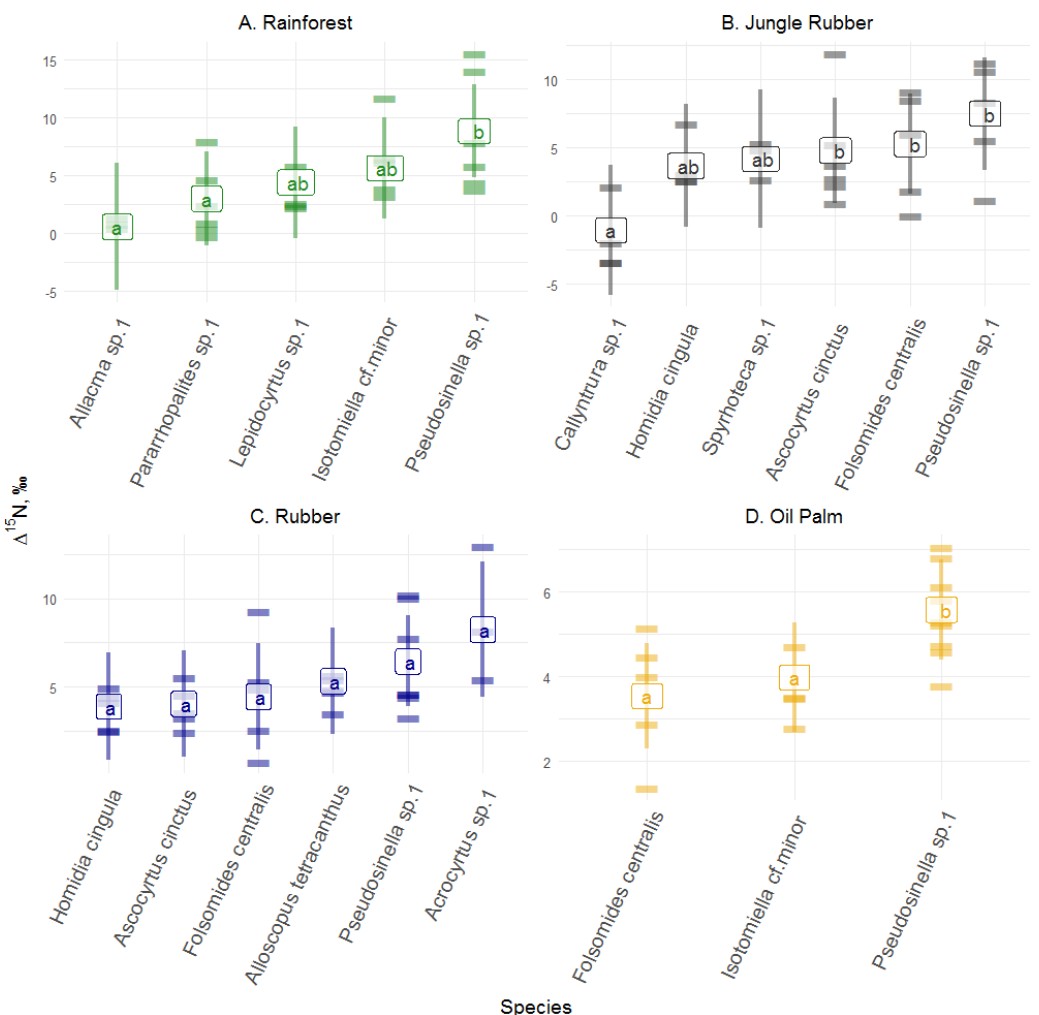

**Figure 3** **Differences of $\Delta^{15}$N values of Collembola in different land-use systems (rainforest, jungle rubber, rubber and oil palm plantations).** Differences of $\Delta^{15}$N values of Collembola in different land-use systems: (A) rainforest, (B) jungle rubber, (C) rubber and (D) oil palm plantation; model-estimated means (lsmeans) with standard deviation. Horizontal stripes represent individual measurements. Only species with more than three replicates per land-use system were analyzed (see Methods). Isotope values of each species in each land-use system with the same letter are not significantly different according to Tukey's honestly significant difference test ($p > 0.05$).

## Differences among Collembola life forms and families

The $\Delta^{15}$N values of Collembola varied among life forms ($F_{24,126} = 1.82, p = 0.018$), but this was not the case for $\Delta^{13}$C values ($F_{24,126} = 1.19, p = 0.260$), with the interaction between life form and land-use system neither being significant for $\Delta^{15}$N ($F_{9,132} = 0.87, p = 0.556$) nor for $\Delta^{13}$C ($F_{9,135} = 0.79, p = 0.619$). Differences between life forms were more pronounced in rainforest and jungle rubber and less in rubber and oil palm plantations (Fig. 4). Euedaphic species were generally most enriched in $^{15}$N, whereas atmobiotic species on average occupied the lowest trophic position (except in rubber plantations), often having $\Delta^{15}$N values below 5.0‰, with epedaphic and hemiedaphic species being intermediate.

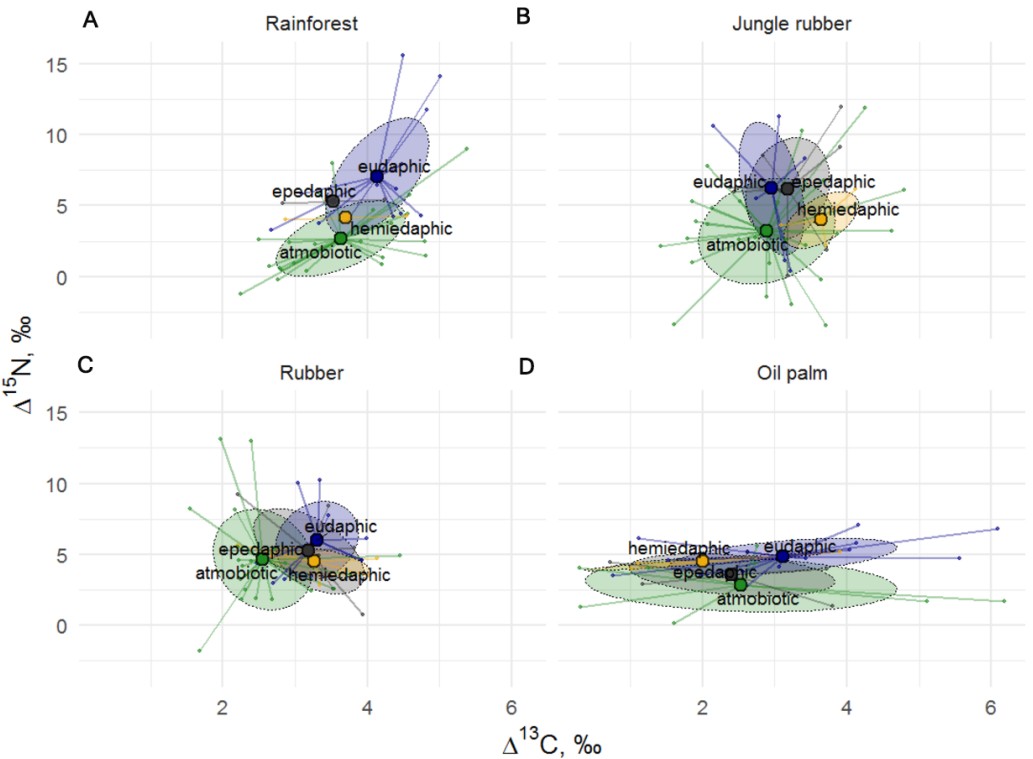

**Figure 4 Stable isotope niches of four Collembola life forms in rainforest, jungle rubber, rubber, and oil palm plantations.** Stable isotope niches of four Collembola life forms in (A) rainforest, (B) jungle rubber, (C) rubber, and (D) oil palm plantations. Ellipses denote 60% confidence intervals, different life forms are shown in color (eudaphic = blue, epedaphic = grey, hemiedaphic = yellow, atmobiotic = green). Large colored dots represent means of land-use systems, small points represent individual measurements.

Similar to results of the life form analysis, the $\Delta^{15}$N ($F_{24,129} = 2.19$, $p = 0.002$) but not $\Delta^{13}$C values ($F_{24,128} = 1.01$, $p = 0.470$) of Collembola varied significantly among families (Fig. 5), with the interaction between family and land-use system neither being significant for $\Delta^{15}$N ($F_{8,122} = 0.34$, $p = 0.945$) nor for $\Delta^{13}$C ($F_{8,121} = 1.84$, $p = 0.075$). Isotomidae and Entomobryidae occupied the highest trophic positions across land-use systems, Symphypleona occupied the lowest trophic position in rainforest and oil palm plantations, whereas in jungle rubber and rubber plantations the lowest trophic position was occupied by Paronellidae. Family and life form identity explained approximately two times less variation in $\Delta^{15}$N values than species identity with $R^2 = 0.29$ in models based on species, and $R^2 = 0.13$ in both models based on families and life forms.

## DISCUSSION

### Variations in trophic niches with land-use system
Results of our study indicate that rainforest conversion into agricultural plantations is associated with changes in basal resources ($\Delta^{13}$C values) of Collembola, but does not significantly affect their average trophic positions ($\Delta^{15}$N values). These findings are in line

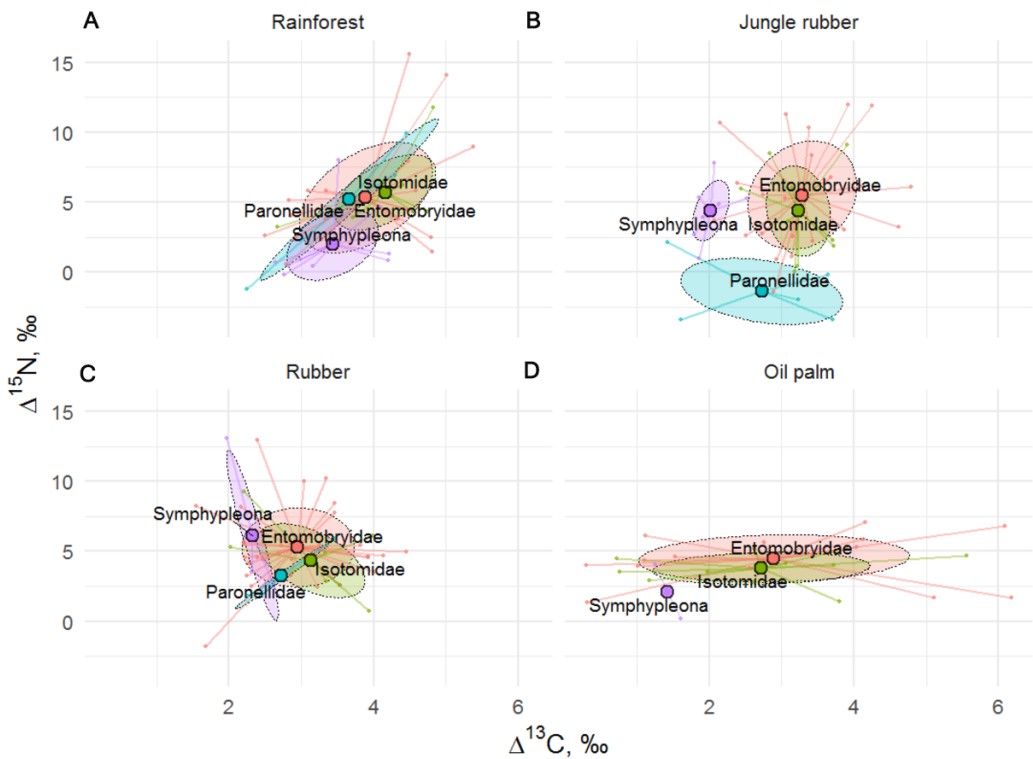

**Figure 5  Stable isotope niche of Collembola families in rainforest, jungle rubber, rubber, and oil palm plantations.** Stable isotope niche of Collembola families in (A) rainforest, (B) jungle rubber, (C) rubber, and (D) oil palm plantations. Ellipses denote 60% intervals, different families are shown in color (Paronellidae = blue, Isotomidae = green, Symphypleona = purple, Entomobryidae = red; Symphypleona comprises the families Sminthuridae, Sminthirididae, Dycirtomidae and Katiannidae). Large colored dots represent means of land-use systems, small points represent individual measurements.

with the results of the study of *Krause et al. (2019)* investigating oribatid mites at the same study sites and showing that the shift in trophic niches to be mainly due to changes in the use of basal resources rather than trophic levels. Similar to previous studies on centipedes, oribatid mites and other soil animal taxa (*Klarner et al., 2017*; *Susanti et al., 2019*; *Krause et al., 2019*), we also found the conversion of rainforest into plantations to be associated in Collembola with a shift from detritivory towards herbivory (i.e., lower $^{13}C$ enrichment). In rainforest and jungle rubber, $\delta^{13}C$ values of Collembola were 2.0–5.0‰ higher compared to leaf litter, which resembles the shift observed in temperate forest ecosystems (*Pollierer et al., 2009*). This "detrital shift" in $\delta^{13}C$ (*Potapov, Tiunov & Scheu, 2019*) presumably is due to acquiring C from saprotrophic fungi and bacteria (*Potapov et al., 2013*) that preferentially use $^{13}C$-rich plant compounds (*Pollierer et al., 2009*). High $^{13}C$ enrichment in most of the studied Collembola species in more natural ecosystems (rainforest and jungle rubber) suggests that they predominantly rely on microorganisms decomposing organic matter rather than on fresh plant material as food source (*Potapov, Tiunov & Scheu, 2019*). Overall, similar enrichment in $^{13}C$ and $^{15}N$ in Collembola in temperate and tropical

ecosystems suggest that Collembola rely little on fresh plant material or mycorrhizal fungi in both, and this may well apply to soil food webs in general.

As indicated by $\Delta^{13}$C values, Collembola species in oil palm plantations used food resources of a wide range of $^{13}$C values, which was not the case in the other land-use systems studied. Variations in $^{13}$C values of food resources in oil palm plantations may be attributed to the variety of management practices of the smallholder systems studied (*Clough et al., 2016*). Potentially, variations in organic inputs due to differences in weeding, herbicide, and fertilization practices, resulted in an overestimation of the trophic niche width in oil palm plantations.

The narrow range of $\Delta^{15}$N values in oil palm plantations suggests similar trophic positions of Collembola species in this land-use system. Notably, the narrow range was due to both the lack of high trophic level (predators, scavengers) and low trophic level species (primary decomposers, specialized lichen feeders). This may reflect the dominance of generalist species in the disturbed habitat of oil palm plantations (*Korotkevich et al., 2018*). Nematode communities also indicated oil palm plantations to be the most disturbed of the four land-use systems studied (*Krashevska et al., 2019*). Further, *Klarner et al. (2017)* found $\Delta^{15}$N values of centipede species to be lower in oil palm compared to jungle rubber and rubber plantations suggesting that trophic chains in oil palm plantations are shorter than in the other land-use systems studied. Overall, these findings suggest that the structure of soil food-webs in monoculture plantations, especially oil palm, is simplified due to reduced number of trophic levels.

The most dominant Collembola species at our study sites, *Pseudosinella* sp.1, occupied a similar trophic niche in each of the land-use systems studied, suggesting that its diet changes little with the conversion of rainforest into plantation systems. *Pseudosinella* sp.1 colonized both litter and soil, and, as indicated by $\Delta^{15}$N values, occupied the highest trophic position among all Collembola species studied. High $\Delta^{15}$N values suggest that this species may either feed on mycrorrhizal fungi, which are enriched in $^{15}$N (*Hobbie, Weber & Trappe, 2001*; *Potapov & Tiunov, 2016*), and/or live as predators feeding e.g., on nematodes, which are similarly abundant across the land-use systems studied (*Krashevska et al., 2019*). Conversion of rainforest may have less affected the mineral soil- and root-based resources than litter resources, and this may explain the high abundance of euedaphic *Pseudosinella* sp.1 in plantations. Similarly, *Krause et al. (2019)* found the trophic niche of dominant species of oribatid mites to change little with the conversion of rainforest into plantations. This suggests that trophic niches of certain species may be little affected by land-use change despite strong changes in the overall food-web structure.

## Trophic differentiation among species

Variations in $\Delta^{13}$C values among Collembola species were more pronounced in rainforest and jungle rubber than in rubber and oil palm plantations. *Korotkevich et al. (2018)* also found the interspecific (in contrast to intraspecific) variation in trophic niches of Collembola to be higher in natural (forest and meadows) than in disturbed habitats (pastures and lawns). Among the Collembola species studied $\Delta^{13}$C values of *Pseudosinella* sp.1 in rainforest, and *Callyntrura* sp.1 and *Sphyroteca* sp.1 in jungle rubber differed

significantly from those of other Collembola species indicating that these species are able to exploit resources not available to the other Collembola species in the respective land-use system.

Similar to $\Delta^{13}$C values, Collembola species in each land-use system, except rubber plantations, also differed in $\Delta^{15}$N values. In jungle rubber, *Callyntrura* sp.1 was most depleted in $\delta^{15}$N among the studied Collembola species indicating that this species occupied the lowest trophic position pointing to phycophagy (*Potapov Korotkevich & Tiunov, 2018*). In rainforest, *Pseudosinella* sp.1 had the highest $\delta^{15}$N values followed by *Isotomiella* cf. *minor* and *Acrocyrtus* sp. suggesting that these species are as microbivores in undisturbed ecosystems. Overall, based on $\Delta^{13}$C and $\Delta^{15}$N values, trophic niche differentiation among species was most pronounced in rainforest, presumably due to the availability of a wider spectrum of food resources and more stable environmental conditions than in plantations. This is likely to result in more efficient food-web functioning in natural ecosystems due to species complementarity (*Loreau & Hector, 2001*), which is partly lost in plantation systems.

## Variations in isotopic niches of Collembola taxa and life forms with land-use systems

Collembola taxa such as Symphypleona and Paronellidae typically had low $\Delta^{15}$N values, indicating that these taxa feed on algae or lichens (*Chahartaghi et al., 2005*; *Potapov Korotkevich & Tiunov, 2018*). Symphypleona as well as Paronellidae are well adapted to aboveground life, both are large-sized and possess well-developed visual systems. Such 'atmobiotic' Collembola have been assumed to live at least in part as herbivores on vascular plants or algae (*Rusek, 2007*). This suggestion is in line with studies from temperate forests investigating variations in stable isotope ratios in Symphypleona (*Chahartaghi et al., 2005*; *Potapov et al., 2016*). Paronellidae predominantly occur in tropical regions also living above the ground suggesting that the microhabitat they live in defines feeding preferences of Collembola across different phylogenetic lineages (see life form discussion below). Entomobryidae and Isotomidae occupied high trophic positions and this also is consistent with earlier studies based on variations in stable isotope ratios suggesting that they predominantly feed on microorganisms colonizing decomposing litter materials (*Chahartaghi et al., 2005*; *Potapov et al., 2016*). Some species of these two families occupied very high trophic positions resembling those of Onychiuridae and Neanuridae in temperate ecosystems suggesting that at least in part they live as predators or scavengers, presumably including nematodes as prey (*Heidemann et al., 2014*). Since Onychiuridae and Neanuridae were rare at our study sites, species from other families might have been able to occupy their trophic niches.

Trophic niches of Collembola in our study varied with life form as shown previously for temperate ecosystems (*Ponge, 2000*; *Potapov et al., 2016*). Conform to the patterns in Collembola families, the results suggest that atmobiotic and epedaphic species occupied the lowest trophic position across land-use systems, whereas euedaphic species such as *Pseudosinella* sp.1 occupied the highest trophic position. This is in line with the results of the study of *Potapov et al. (2016)* indicating that species inhabiting deeper soil layers

(hemiedaphic and euedaphic) are more enriched in $^{15}$N than those living in litter and above the ground (epedaphic and atmobiotic). High $\delta^{15}$N values may result from feeding on ectomycorrhizal fungi which are enriched in $^{15}$N (see above; *Hobbie, Weber & Trappe, 2001*; *Potapov & Tiunov, 2016*), however, this unlikely applies to tropical forests where trees predominantly form mutualistic interactions with arbuscular mycorrhizal fungi. We attributed low $\delta^{15}$N values in epedaphic and atmobiotic Collembola to algae or lichen feeding, which is widespread in Collembola in temperate forests (*Potapov Korotkevich & Tiunov, 2018*), but in our study only few species had $\delta^{15}$N values below those of litter. This contradicts results based on fatty acid analysis suggesting that Collembola feed more on algae in tropical than in temperate ecosystems (*Susanti et al., 2019*). To clarify the contribution of algae in soil food webs in tropical and temperate ecosystems, more data on stable isotope composition of various food resources in tropical forests, or direct experimentation, are needed. Differences between life forms were more pronounced in rainforest and jungle rubber and less in plantation systems, which may reflect the more pronounced litter layer in the former than the latter (*Krashevska et al., 2019*). Overall, the results suggest that similar to oribatid mites (*Tsurikov, Ermilov & Tiunov, 2019*) the trophic niche structure in Collembola communities in temperate and tropical forests is generally similar and this is partly explained by taxonomic affiliation and life form.

## CONCLUSION

We showed that the conversion of rainforest into agricultural plantations, such as rubber and oil palm, is associated with changes in trophic niches of Collembola. The use of food resources shifted towards herbivory, with the range of food resources of Collembola in oil palm plantations being the highest, likely due to the heterogeneity in management. By contrast, the range of trophic positions in oil palm plantations was low suggesting that the trophic structure is simplified lacking high but also low trophic levels. This is further supported by the less pronounced trophic niche differentiation among species in monoculture plantations. Similar to the pattern in oribatid mites (*Tsurikov, Ermilov & Tiunov, 2019*), the structure of trophic niches in tropical Collembola communities resembled that in temperate forests. Life form and family identity explained about half of the species-level variation; atmobiotic species occupied the lowest and euedaphic species the highest trophic position, but the difference was less pronounced in plantations. Overall, the results document that changes in community composition associated with the conversion of rainforest into plantation systems are followed by shifts in the trophic structure and trophic niches in Collembola communities, potentially compromising ecosystem functions and food-web stability in plantations.

## ACKNOWLEDGEMENTS

We thank the State Ministry of Research and Technology of Indonesia (RISTEK), the Indonesian Institute of Sciences (LIPI), Ministry of Forestry (PHKA) and Restoration Ecosystem Indonesia Harapan.

## Funding

This study was funded by the German Research Foundation (DFG)—project number 192626868—in the framework of the collaborative German-Indonesian research project CRC990. The funders had no role in study design, data collection and analysis, decision to publish, or preparation of the manuscript.

## Grant Disclosures

The following grant information was disclosed by the authors:
German Research Foundation (DFG): 192626868.
Indonesian Research Project: CRC990.

## Competing Interests

The authors declare there are no competing interests.

## Author Contributions

- Winda Ika Susanti performed the experiments, analyzed the data, prepared figures and/or tables, authored or reviewed drafts of the paper, and approved the final draft.
- Rahayu Widyastuti conceived and designed the experiments, authored or reviewed drafts of the paper, contributed to organize research permition, and approved the final draft.
- Stefan Scheu conceived and designed the experiments, authored or reviewed drafts of the paper, and approved the final draft.
- Anton Potapov conceived and designed the experiments, performed the experiments, authored or reviewed drafts of the paper, and approved the final draft.

## Field Study Permissions

The following information was supplied relating to field study approvals (i.e., approving body and any reference numbers):

Field collection was conducted under the research permit No. 389/SIP/FRP/SM/X/2013 issued by the State Ministry of Research and Technology of Indonesia (RISTEK) with collection permit No. S.07/KKH-2/2013 issued by the Ministry of Forestry (PHKA). The following persons and organizations for granted us access to and use of their properties: village leaders, local plot owners, PT Humusindo, PT Perkebunan Nusantara VI, Harapan Rainforest, and Bukit Duabelas National Park.

## Data Availability

Raw data are available in Supplementary Files.

## Supplemental Information

Supplemental information for this article can be found online at http://dx.doi.org/10.7717/peerj.10971#supplemental-information.

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
