# Peer review of "Trophic niche differentiation and utilisation of food resources in Collembola is altered by rainforest conversion to plantation systems"

_PeerJ, doi:10.7717/peerj.10971_

## Round 0.1 · original submission · Minor Revisions

Three reviewers as experts in their field have reviewed your manuscript and proposed a number of changes which you should strongly follow in your revision. I look forward to receive your revised manuscript.

Reviewer 1 ·

Basic reporting

The manuscript written by Susanti et al. studied trophic niche differentiation of Collembola in different forest types in Indonesia. The manuscript is well-written and can contribute to the trophic ecology of soil organisms.
Citing papers are enough, but need to be confirmed. E.g. Potanov et al. 2019 was cited in the text, but not found in references. Instead, Potanov et al. 2018, 2018a, and 2018b were found in references, which need to be identified. There are some other mistakes in forms, like (Korotkevich, Potapov, Tiunov, & Kuznetsova, 2018), (Josef Rusek, 1998) etc.
There are some small corrections needed as isotopic point of view (see General comments).

Experimental design

The manuscript used the study system that has already been used in several papers. The results are nearly the same as the previous papers, and the results seem to be rigorous.
Cite references to show the validity of using glycerol:water solution and 70% ethanol for the preparation of isotope measurement.

Validity of the findings

The authors studied Collembola species/genera among the vegetations. The isotope values are generally understandable among the vegetations. Results provide interesting information on tropical ecology.

Additional comments

There are some minor comments to the manuscript.
Line 170 The factor 1000 is an extraneous numerical factor and should be deleted (Coplen, TB (2011) Rapid Commun. Mass Spectrom. 25: 2538–2560).
Line 172 Vienna PD Belemnite is a strange expression. I suggest either Vienna Pee Dee Belemnite or VPDB.
Line 173 Multiple working standards should be used to calibrate isotope values (Skrzypek (2013) Anal Bioanal Chem 405: 2815-2823).
Lines 216, 314 and several places. The authors should not use “enriched” in “δ13C” of “δ15N”. Instead, I suggest either “All Collembola species were enriched in 13C relative to leaf litter…” or “All Collembola species had higherδ13C values relative to leaf litter… (The authors used correctly in line 276 and several places.)

Reviewer 2 ·

Basic reporting

The manuscript was well structured in clear English. Introduction may need to be revised. I checked the raw data was supplied. Figures seemed sufficient.

I agree with the authors’ claim that there are few studies investigating Collembola food attributes in tropical ecosystems. However, I could not understand what is the difference of soil ecosystem between Tropical and Temperate forests. Is there some difference of the soil systems for Collembola between tropical rain-forests and temperate forests? Please explain it.

Are there some problems of plantations in tropical forests, in ecological or economical sense?

Experimental design

Do you have species abundance data of the community?
Did the measured species number in Figure2-3 reflect to real species richness or community compositions?
Stable C-N isotope mapping at community level should have been biased in species selection processes, although the authors explained that species which were measured isotopes were covered 70% individuals (L159). If authors have the abundance data in each land-use, you can avoid the bias because of the direct evidence. Table of each plot only with dominant species abundance and total number may be sufficient for this purpose.

L130: Did herbicides directly decline some soil Collembola?

I think that pesticide, herbicides or fertilizer in management forests should disturb the C-N isotope ratios. Please explain it. (see also L323)

L183-207: Authors used the statistical analyses in three LMM models for land-use, species, taxonomy and lifefrom. Which was the most important factor for stable C-N isotope ratios?
I think that species identity should not be used as random effect in the first analysis, because species effect on stable isotope ratio should be important for the community isotope ratios.

L192: The range of delta C and N should be quantitatively evaluated using some criteria, such as the difference between minimum-maximum, functional diversity index (such as Rao’s quadratic entropy) or coefficient of variance (C.V.).

Validity of the findings

Interpretation of the results were described in intrinsic reasons, such as taxonomic difference and life-form difference in Collembola to land-use. However, as Figure 4 & 5, lifeform mapping and taxonomic mapping were not consistent with each other land-use type, and seemed not so robust the discussion about life-form and taxonomy in line of Potapov et al. (2016 Soil Biol Biochem).
I think that the changes by land-use in organic matter composition as habitat structure for Collembola, such as L, F, H compositions, may be important for Collembola community. The variations of isotopes in taxonomy and life-form (Potapov et al. 2016 Soil Biol Biochem) may be context dependent. Please explain how the isotope variations changed by land-use in regards to life-form and taxonomy.
Collembola food items, represented to C-N isotopes, partly reflect to the humification gradients of the habitat substrate (Hishi et al., 2007 Soil Biol. Biochem), suggesting organic matter accumulation may relate with the ranges of the Collembola feeding items. Therefore, the ranges of C-N isotopes might be due to the range of humus gradient. If authors have the data, try to explain the variations of isotopes from accumulation of forest floor of each land-use.

Overall, data of this study of tropical Collembola was well structured, and useful for broad readers.

Additional comments

I agree with the authors’ claim that there are few studies investigating Collembola food attributes in tropical ecosystems. However, I could not understand what is the difference of soil ecosystem between Tropical and Temperate forests. Is there some difference in soil system for Collembola in tropical rain-forest?

Reviewer 3 ·

Basic reporting

Overall the manuscript is well written, in clear and professional English throughout - for minor corrections see general comments below. The article is well structured, figure 1 took me a while to understand the interpretation, but that is because it is a violin plot rather than a normal box plot - I do not think the authors should change it, but it might be worthwhile to include a more in-depth description of how the figure is to be interpreted, so greater understanding can be drawn without resorting to google! Figures 2 and 3 are poor picture quality and a little blurry - this is likely due to a low resolution inclusion when uploading, but again should be checked in proofs. Also the order of the individual species should be the same position across the landuses - Pseudosinella for example should be at the end for all, as in all habitats.

Experimental design

Experimental design is rigorous and suitable for publication. More emphasis should be included on the number of treatments (land-uses) within the abstract, as well as replication within the results/figures.

However, I am unsure why only "dominant" species were chosen for presentation of isotopic analysis - is this due to biomass of samples? Looking at the raw data, it looks like all Collembola had their isotopic composition assessed?

I also think the results would be strengthened if abundance of Collembola was included - the numbers are there in the raw data. What is the difference between the different landuses - there looks like there is a similar abundance, this is interesting and could be discussed. Also looking at the individual species may provide interesting discussion point - there are quite large differences in abundance between the different landuses for Pseudosinella for example.

Did species found in different habitat (e.g. Pseudosinella) have different isotopic results between habitats - any significant differences? I didn't see this stated in results.

Validity of the findings

This is an interesting and novel study that looks at tropical locations for isotopic analysis of Collembola species, with comparison to temperate systems, and other soil fauna. Although I am surprised they did not consider the work of Illig that focused on tropical montane rainforests and isotopic composition of the soil fauna (or the tropical papers that have cited this)?

The differences in Collembola life forms was a novel way of analyzing the data and provided a greater understanding of habitat differences. It was well described within the text.

Additional comments

A useful study that provides additional information regarding the trophic structure of the soil food web, focusing on Collembola within different habitats in the tropics.

Minor comments are as follows:
Abstract - it is not clear how long after conversion the oil and rubber plantations have been growing. Also it is unclear that there are four landuses being investigated.

line 58 - life not live
line 81-82 Further, food resources and trophic levels also vary among taxa, for example, Poduromorpha...
line 85 - sparse not virtually lacking
line 97 - add monocultures of after and and before rubber and oil palm

---

## Round 0.2 · Minor Revisions

As indicated by reviewer 3, there are still some minor points to be resolved, specifically the re-ordering of species in Figs. 2 and 3 and the uptaker of the explanation for use of "dominant species" in the revised manuscript.

Reviewer 2 ·

Basic reporting

no comment

Experimental design

no comment

Validity of the findings

no comment

Additional comments

I have checked that authors properly responded to my comments. I have agreed with their explanations.

Reviewer 3 ·

Basic reporting

In response to the authors statements regarding earlier reviews comments:
They have completed the majority of changes suggested by reviewers. However I am unsure why they did not reorder the species of Collembola in figures 2 and 3 - these figures are supposed to indicate the variation in isotopes of the different species of springtail across the different habitats. The order they have the species in "based on mean values across land-use systems" highlights that all land-use systems have springtails with a similar range of isotopic compositions. Whereas keeping each species in the same position on the graph for each of the land-uses would indicate the difference in isotopic composition for that species.

Experimental design

In the rebuttal letter the explanation for use of "dominant species" is clear, understandable and a valid reason, however this is not made clear within the manuscript.

Validity of the findings

Surprised they have separated abundance and isotopic composition into two papers, rather than one strong one. The authors state they have included abundance data in the supplementary data, this doesn't really add to the paper if it's not going to be discussed within the paper alongside the isotopic results.

Additional comments

Minor comments are as follows:

line 122 first mention of Latin names of rubber / oil plantations, should be included in abstract and earlier in MS
line 139 use the chemical name for the product rather than the commercial
line 151-154 are acknowledgements not methods and should be deleted
line 372 typo

---

## Round 0.3 · accepted · Accept

Thank you very much for the revision of the manuscript and the consideration of the reviewers' proposals.